# Prevalence of Post-Traumatic Stress Disorder (PTSD) in Healthcare Workers following the First SARS-CoV Epidemic of 2003: A Systematic Review and Meta-Analysis

**DOI:** 10.3390/ijerph192013069

**Published:** 2022-10-11

**Authors:** Bastien Alberque, Catherine Laporte, Laurie Mondillon, Julien S. Baker, Martial Mermillod, George Brousse, Ukadike Chris Ugbolube, Reza Bagheri, Jean-Baptiste Bouillon-Minois, Frédéric Dutheil

**Affiliations:** 1Université Clermont Auvergne, CNRS, LaPSCo, Physiological and Psychosocial Stress, CHU Clermont-Ferrand, 63000 Clermont-Ferrand, France; 2Université Clermont Auvergne, Clermont Auvergne INP, CHU Clermont-Ferrand, CNRS, Institut Pascal, 63000 Clermont-Ferrand, France; 3Centre for Health and Exercise Science Research, Hong Kong Baptist University, Kowloon Tong, Hong Kong 999077, China; 4Université Grenoble Alpes, CNRS, LPNC, 38100 Grenoble, France; 5Département de Psychiatrie, Université Clermont Auvergne, CHU Clermont-Ferrand, EA 7280 Clermont-Ferrand, France; 6School of Health and Life Sciences, University of the West of Scotland, South Lanarkshire G72 0LH, UK; 7Department of Exercise Physiology, University of Isfahan, Isfahan 81746-73441, Iran

**Keywords:** public health, mental health, occupation, infection, predictive strategy

## Abstract

The world is still in the grip of the severe acute respiratory syndrome coronavirus 2 (SARS-CoV-2) pandemic, with putative psychological consequences for healthcare workers (HCWs). Exploring the prevalence of post-traumatic stress disorder (PTSD) during the first SARS-CoV-1 epidemic in 2003 may inform us of the long-term effects of the actual pandemic, as well as putative influencing factors such as contact with the virus, time effects, or the importance of some sociodemographic data. This information may help us develop efficient preventive strategies. Therefore, we conducted a systematic review and meta-analysis on the prevalence of PTSD in HCWs following the SARS-CoV-1 in 2003. PubMed, Embase, Google Scholar, Psychinfo, and Web of Science were searched until September 2022. Random-effects meta-analyses were stratified by the time of follow-up. We included 14 studies: 4842 HCWs (32.0 years old, 84% women). The overall prevalence of PTSD was 14% (95CI 10 to 17%). The prevalence of PTSD was 16% (8 to 24%) during the epidemic, 19% (16 to 22%) within 6 months after the epidemic, and 8% (4 to 13%) more than one year after the end of the epidemic. The longest follow-up was three years after the epidemic, with 10% of HCWs with PTSD. Nevertheless, the prevalence of PTSD was significantly lower more than one year after the end of the epidemic than the first six months after the epidemic (Coefficient −10.4, 95CI −17.6 to −3.2, *p* = 0.007). In conclusion, the prevalence of PTSD in HCWs was high during the first epidemic of SARS-CoV in 2003 and remained high in the long term. The lessons from the SARS-CoV-1 epidemic may help prevent a wave of PTSD following the latest COVID-19 pandemic.

## 1. Introduction

Post-traumatic stress disorder (PTSD) is a mental disorder that may develop after exceptional threatening events [1]. PTSD can occur after a single traumatic event or prolonged exposure to trauma [1]. Predicting who will develop PTSD is a challenge [2]. PTSD is a public health problem, increasing the risk of suicide and other psychiatric disorders [1,2,3]. The prevalence of PTSD in the general population range from 2 to 8% depending on studies [1,2]. In healthcare workers (HCWs), the prevalence of PTSD has been described following traumatic events such as earthquakes [3] or terrorist attacks [4], but HCWs can also be at risk from highly severe and contagious diseases [5]. Indeed, HCWs are on the front line in the care of patients in any epidemic [6] and are thus at risk of PTSD [7]. The current context of the global pandemic caused by SARS-CoV-2 is putting health staff to the test physically and psychologically [6]. A tsunami of PTSD may follow the SARS-CoV-2 pandemic [8]. Exploring the prevalence of PTSD following previous epidemics may inform us of the psychological consequences of the SARS-CoV-2 pandemic [9,10,11] and may help to build an efficient preventive strategy. Moreover, since we are still in the SARS-CoV-2 pandemic, learning from the long-term consequences of the previous epidemics seems salient. In 2003, the first SARS-CoV epidemic limited its spread mostly to Southeast Asia and Canada [12]. Despite some studies attempting to describe PTSD in the HCWs who took care of patients during the first SARS-CoV epidemic [13,14,15], no robust conclusions were drawn on the prevalence. Several questionnaires have been proposed to assess the symptoms of PTSD, such as the Impact Event Scale (IES) [16,17] or its latest revised edition (IES-R) [18] and the Distress Tolerance Scale (DTS) [19,20]. Despite not being demonstrated in HCWs, the occurrence of PTSD has been shown to increase in the general population over the first year following a stressful event, probably because people suffer from re-experiencing and avoidance symptoms [21]. PTSD can also be a chronic disease that evolves with time and treatment [1,2]; however, the long-term consequences of the first epidemic of SARS-CoV are under debate. A previous meta-analysis was conducted on the psychological consequences of SARS-CoV-1 but did not assess the prevalence of PTSD and did not study putative influencing factors, such as contact with the virus, time effect, or the importance of some sociodemographics [22]. Although never demonstrated in HCWs in contact with SARS-CoV-1 patients, it seems that dose–response can be applied to PTSD [23,24]. Some sociodemographic data also happen to be predictive of people at risk of PTSD, such as women [1,2,25]. Considering all of the aforementioned factors, the risk of HCWs experiencing PTSD during an epidemic is not well known.

Therefore, we aimed to conduct a systematic review of the literature and meta-analysis on the prevalence of PTSD in HCWs. We further aimed to assess if the time from the epidemic impacted the prevalence, to determine whether direct contact with SARS-CoV-1 patients affected the prevalence of PTSD, and to identify the sociodemographic data that may contribute to the risk of PTSD in HCWs.

## 2. Materials and Methods

### 2.1. Literature Search

We reviewed all of the studies involving psychological and psychiatric issues in HCWs following the first SARS-CoV epidemic of 2003. The main databases (PubMed, Embase, Psychinfo, Web of Science, and Google Scholar) were searched using the following keywords: («Health care workers» OR “Healthcare workers”) AND («Psychological disorder» OR «Psychiatric disorder») AND («SARS-CoV» NOT «SARS-CoV-2» NOT «COVID») until September 2022 (details of the search strategy are listed in Appendix A). To be included, the studies had to describe our main primary outcome, i.e., to describe PTSD in HCW populations during or following the first SARS-CoV epidemic of 2003. We excluded studies that assessed PTSD in patients or other psychological disorders. All articles compatible with our inclusion criteria were included independently of the year of publication. We limited our search to articles written in English. We imposed no limitation on regional origin. Studies needed to be primary research. In addition, reference lists from all publications meeting our inclusion criteria were manually searched to identify any further studies that were not found with the electronic search. Ancestry searches were also completed on previous reviews to locate other potentially eligible primary studies. The search strategy is presented in Figure 1. Two authors (Bastien Alberque and Reza Bagheri) conducted the literature searches, reviewed the abstracts and articles independently, checked suitability for inclusion, and extracted the data. When necessary, disagreements were solved with a third author (Frédéric Dutheil).

### 2.2. Data Extraction

The primary endpoint was the analysis of PTSD in the HCW population during the SARS-CoV epidemic of 2003. The data collected included the first author’s name, publication year, study design, main and secondary outcomes of each study, as well as the inclusion and exclusion criteria, sociodemographic (sample size of HCWs, age, percentage of males, marital status, number of HCWs having children, seniority and occupation), and the measures and prevalence of PTSD (number of PTSD, tool assessment and related scores, time from the epidemic, working in a department with high or low contact with patients infected by SARS-CoV-1).

### 2.3. Quality of Assessment

We used two grids to check the quality of the included articles [25,26]. For the Newcastle–Ottawa Scale (NOS), each item was assigned a judgment of “Yes” (1 point), “No” (0 point), or “Cannot say” (0 point) (Figure 2 and Appendix B). We also used the “Strengthening the Reporting of Observational studies in Epidemiology” (STROBE—32 items/sub-items) [27,28]. The STROBE Statement is a checklist related to the title, abstract, introduction, methods, results, and discussion sections of articles (Appendix B). For both the NOS and STROBE checklists, we attributed one point per item or sub-item, then converted it into an overall percentage reflecting the quality of each article.

### 2.4. Statistical Considerations

We used Stata software (v16, StataCorp, College Station, TX, USA) for the statistical analysis. The main characteristics were synthetized for each study population and reported as mean ± standard deviation (SD) for continuous variables and number (%) for categorical variables. When the data could be pooled, we conducted random effects meta-analyses (DerSimonian and Laird approach) to assess the prevalence of PTSD in HCWs [29]. We stratified our analysis based on both clinical relevancy and the available and sufficient data for each stratification, i.e., we stratified our results by time from the epidemic (during the epidemic, 1 to 6 months after the end of the epidemic, and more than one year after), and by contact (low and high) of HCWs with patients infected by SARS-CoV-1. We evaluated heterogeneity in the study results by examining forest plots, confidence intervals (CI), and I-squared (I^2^). I^2^ is the most common metric to measure heterogeneity between studies, ranging from 0 to 100%. Heterogeneity is considered low for I^2^ < 25%, modest for 25 < I^2^ < 50%, and high for I^2^ > 50%. Low heterogeneity is often considered a good quality indicator. A high heterogeneity may signify variability between the characteristics of the included studies, such as differences in cut-offs for the diagnosis of PTSD or differences in the sociodemographic results of HCWs (age, gender, departments, etc.). We also searched for potential publication bias by examining the funnel plots of these meta-analyses. We verified the strength of our results by conducting further meta-analyses after the exclusion of studies that were not evenly distributed around the base of the funnel. When possible (sufficient sample size), meta-regressions were proposed to study the relationship between the prevalence of PTSD and the time from the epidemic, contact with SARS-CoV-1 patients, and sociodemographic (age, gender, marital status, children, seniority, and occupation). Particular attention was paid to the scale used to assess the prevalence and symptoms of PTSD. The results were expressed as regression coefficients and 95% CI. *p*-values of less than 0.05 were considered statistically significant. Finally, because cut-offs for the diagnosis of PTSD may differ between the included studies, we repeated all meta-analyses and meta-regressions on scores of PTSD for each dimension of the IES: IES-total, IES-intrusive, and IES-avoidance.

## 3. Results

An initial search produced a possible 1165 articles (Figure 1). Removal of duplicates and use of the selection criteria reduced the number of articles reporting PTSD in HCWs to 14 articles for the systematic review [13,14,15,30,31,32,33,34,35,36,37,38,39,40] and 13 articles for the meta-analysis—we excluded one study because there was no number or scores of PTSD [35]. All of the included articles were written in English. The descriptive characteristics of the included articles are summarized in Table 1.

### 3.1. Study Designs of Included Studies

The included studies were published from 2004 [33,35,40,41] to 2009 [13]. All of the studies were cross-sectional except for two cohort studies [15,34], and all of the studies were conducted in South East Asia except for two studies in Canada [15,35].

### 3.2. Quality of Articles

Using the NOS criteria, the quality of the included studies was good, with a mean score of 75.0 ± 28.0%, ranging from 25% [39] to 100% [14,16,32,33,38,40], thus achieving heterogeneity between the studies. They performed worst in controlling for confounders, as well as in the lack of details surrounding the exposure (Figure 2 and Appendix B). Using STROBE, the scores were 62.8 ± 7.2%, ranging from 56.3% [30] to 71.9% [33]. Most studies performed less in transparency regarding addressing missing data, the absence of flow charts, bias descriptions, and the description of statistical analyses (Appendix B). All of the studies mentioned ethical approval except for four [30,31,32,37].

### 3.3. Aims of Included Articles

All of the studies shared the main objective of assessing the psychological or psychiatric impact of the SARS-CoV-1 epidemic of 2003 on HCWs, except for one study that had the main objective of assessing the alcohol use of HCWs exposed to the SARS-CoV-1 outbreak [40]. More specifically, six studies aimed to assess the psychological impact and symptoms of distress [13,14,15,30,31,32,37], four studies assessed the psychiatric morbidity of the 2003 epidemic [35,36,40,42], and one study was on perceived stress during and after the epidemic [36]. Twelve of the thirteen included studies mentioned secondary objectives: to evaluate the impact of the epidemic on the lives of HCWs in six studies [15,32,34,35,39,42], to search for factors associated with distress in five studies [16,33,36,38,40], and to search for a relationship between PTSD and alcohol abuse in one study [40]. In total, only two studies directly named PTSD, but only in their secondary objectives [36,40].

### 3.4. Inclusion and Exclusion Criteria of Included Studies

All of the included studies focused on HCWs and were conducted on a voluntary basis. All of the studies were conducted on hospital grounds, either monocentric [13,14,30,31,32,33,37,38,40] or multicentric [16,36,38], except one study that was conducted outside of a hospital [39]. The occupations included differed between the studies: only nurses [31,38], only doctors [39], nurses and physicians [30,33], and all HCWs with [14,34,39,41] or without administrative staff [15,16,36,38]. Only two studies had exclusion criteria: no comprehension of the questionnaire or no contact with SARS-CoV-1 patients [33] and a psychiatric history or whether HCWs contracted the virus [36].

### 3.5. Population

The sample size ranged from 47 [37] to 1257 [32] HCWs, for a total of 4842 HCWs. Age was reported in all studies. The mean age of HCWs was 32.0 (95CI 29.2 to 34.7). Gender was reported in all studies except two [30,37]. The proportion of men was 16% (95CI 13 to 18%), ranging from 0 [31,38] to 60.6% [39]. Marital status was described in eleven studies [13,14,15,30,31,32,33,35,37,38,40]. HCWs having children or not was described in three studies [16,33,40]. Other parameters were described in a few studies, such as religion [33], ethnicity [32,39,42], education [14,37,38,41], previous psychological distress [34], and seniority [31,32,33,34,38].

### 3.6. Assessment of PTSD

Among the thirteen included studies, eleven studies assessed PTSD with the IES scale [13,15,30,31,32,33,34,36,37,39,40], and two with the DTS scale [14,38]. All of the studies reported the prevalence of HCWs with PTSD, except four that reported only scores using the IES scale [34,38,42].

The **IES** was initially a questionnaire or self-questionnaire of 15 items (IES) that was used in seven of the included studies [15,30,31,32,33,34,37,40], then the revised in a version with 22 items (IES-R) was used in four studies [14,38,41,42]. All of the items were scored on a 5-point Likert scale ranging from 0 to 4 (0 = not at all, 4 = extremely), for a total score ranging from 0 to 60 for the IES [16,17], and from 0 to 88 for the IES-R [18]. The IES assesses two dimensions: the IES intrusive, which represents the ideas that disturb the normal thinking of a person, and the IES avoidance, which represents the actions a person would take to avoid a situation. The IES-R added a third dimension: the IES hyperarousal, which assesses trouble concentrating, anger, irritability, and hypervigilance. The IES score represents the risk of PTSD: the higher the score, the higher the risk. The cut-off for symptomatic PTSD on the IES score is under debate. Consequently, the cut-offs varied between our included studies: IES > 20 [13,40], IES > 25 [15,33], IES > 30 [30,37], and IES > 35 [31]. Four studies did not report the cut-off used [34,36,38,42].

The **DTS** is a self-report 17-item Likert scale assessing the 17 symptoms of PTSD listed in the DSM-IV [19]. All of the items evaluated the frequency and severity of symptoms, each scored on a 5-point Likert scale ranging from 0 to 4 (0 = never/no stress, 4 = every day/extremely stressed), for a total score ranging from 0 to 136 (0 to 68 for severity and 0 to 68 for frequency). Similar to IES, the higher the score, the higher the risk of PTSD. Despite the fact that both the frequency and severity scores can be determined, the two studies that used the DTS scale reported only prevalence [14,38]. They used the Chinese version (DTS-C) that was developed for Chinese-speaking individuals [20], also including 17 items on two Likert scales (frequency/severity) from 0 to 4. The most common cut-off for PTSD is a DTS score of >40/138 [19]. In our meta-analyses, one study used a cut-off of >23 [38] and one >40 [14].

### 3.7. Characteristics of Exposure

The time from the epidemic was reported in all of the studies: during the epidemic in six studies [32,33,37,38,39,40], 1 to 6 months after the epidemic in three studies [15,32,35], and more than one year after the epidemic in five studies [14,16,36,38,41]. The longest follow-up in included studies was 3 years [13,40]. Contact with SARS-CoV-1 was reported in all of the included studies. We defined “high contact” as HCWs who took care of patients affected by SARS-CoV-1 or who were working in the SARS departments, and“low contact” as other HCWs. Twelve studies described HCWs in high contact with the virus [13,14,15,30,31,32,33,34,36,38,39,40], seven studies compared “high contact” and “low contact” HCWs [13,15,30,31,32,38,39], and one study only included “low contact” HCWs [37].

### 3.8. Meta-Analysis of Prevalence of PTSD in HCWs Population

The overall prevalence of PTSD was 14% in HCWs (95CI 10 to 17%). More specifically, the prevalence was 16% (8 to 24%) during the epidemic, 19% (16 to 22%) between 1 to 6 months after the end of the epidemic, and 8% (4 to 13%) more than 1 year after the epidemic. On the overall timeline, HCWs with high contact with the virus had a prevalence of PTSD of 15% (10 to 19%), and those with low contact with SARS-CoV-1 had a prevalence of 12% (7 to 17%). The stratified meta-analysis by time and by contact with SARS-CoV-1 showed a prevalence of PTSD of 22% (11 to 33%) in HCWs with high contact and 9% (2 to 17%) in those with low contact during the epidemic, 18% (14 to 23%) and 19% (16 to 23%) in high- and low-contact HCWs between 1 to 6 months after the end of the epidemic, and 9% (3 to 14%) and 7% (5 to 9%) more than 1 year after the epidemic. Most I^2^ were high (>80%), highlighting the heterogeneity between the results (Figure 3 and Figure 4).

### 3.9. Meta-Analysis of IES Score in HCWs Population

IES total score was 10.5 (95CI 6.4 to 14.6): 20.0 (17.7 to 22.3) in HCWs with high contact and 8.2 (2.9 to 13.6) in HCWs with low contact with the SARS-CoV-1. More specifically, the IES total was 15.1(7.2 to 23) during the epidemic: 20.8 (17.9 to 23.8) in high contact and 7.7 (2.2 to 13.2) in low contact HCWs; and 12.6 (4.4 to 20.8) between one to six months after the epidemic: 18.0 (11.7 to 24.4) in high contact and 15.7 (−5.1 to 36.5) in low contact HCWs.

IES avoidance score was 5.0 (95CI 2.7 to 7.3) over the whole timeline: 8.8 (4.0 to 13.6) in HCWs with high contact and 3.9 (1.3 to 6.5) in HCWs with low contact with the SARS-CoV-1. More specifically, IES avoidance was 6.3 (2.7 to 9.9) during the epidemic: 11.7 (3.7 to 19.7) in high contact and 4.7 (1.0 to 8.5) in low contact HCWs; and 7.4 (2.1 to 12.7) between one to six months after the epidemic: 7.2 (1.2 to 13.1) in high contact and 8.4 (−3.2 to 20.0) in low contact HCWs.

IES intrusive score was 6.2 (95CI 3.7 to 8.7) over the whole timeline: 6.5 (3.0 to 10.1) in HCWs with high contact and 5.9 (2.4 to 9.3) in HCWs with low contact with the SARS-CoV-1. More specifically, IES intrusive was 7.1 (3.8 to 10.4) during the epidemic: 13.2 (5.6 to 20.8) in high contact and 5.7 (2.0 to 9.3) in low contact HCWs; and 4.7 (0.7 to 8.7) between one to six months after the epidemic: 5.1 (1.4 to 8.8) in high contact and 7.2 (−2.7 to 17.2) in low contact HCWs.

Most I^2^ were low (<15%) for IES total, IES avoidance, and IES intrusive. IES arousal was never reported, and other IES scores were never reported more than one year after the end of the epidemic (Appendix C and Appendix D).

### 3.10. Meta-Regression and Influencing Factors

The prevalence of PTSD was lower than one year after the end of the epidemic compared to 1 to 6 months after the epidemic (Coefficient −10.4, 95CI −17.6 to −3.24, *p* = 0.007) and tended to be lower compared to during the epidemic (−6.63, −14.2 to 0.96, *p* = 0.084). HCWs with children were more at risk of PTSD (0.85, 0.27 to1.42, *p* = 0.13). There was no significant effect of contact with the virus (high vs. low) or sociodemographic (age, gender, marital status) (Figure 5). No factors influencing the IES scores (total, avoidance, intrusive) were significant (Appendix E).

## 4. Discussion

The main findings were that the prevalence of PTSD for the first SARS epidemic in 2003 among HCWs was high, around 14%. The prevalence seemed to remain high even more than one year after the end of the epidemic. All of the HCWs were at risk of PTSD and not only those working in the −1 unit. Among personal risk factors, only having children may seem to increase the risk of PTSD.

### 4.1. PTSD in Healthcare Workers: A Public Health Issue

Despite some studies reporting the prevalence of PTSD in the general population, ranging from 2 to 8% [1,2], there is no data to our knowledge on HCWs outside of exceptional traumatic events, such as earthquakes [3] or terrorist attacks [4]. We demonstrated a high prevalence of PTSD in HCWs during the first epidemic of SARS-CoV-1 in 2003. This high prevalence is a problem that needs to be considered, particularly given the actual context of the COVID-19 pandemic [8]. In the current pandemic, HCWs are psychologically and physically exposed to the virus [6]. PTSD could increase the risk of suicide in HCWs who are already at higher risk by their work demands [42]. Some studies also reported an increase in medical errors in HCWs suffering from burnout, suicidal ideation, and PTSD [41,43]. Thus, poorer quality of care may be a consequence of PTSD in HCWs [44]. Moreover, PTSD is also linked with oxidative stress, metabolic disorders, stroke, and cardiovascular disease, even in young HCWs [45,46,47,48]. Since the intensity of the epidemic of SARS-CoV-2 is much larger than SARS-CoV-1, the number of PTSD in HCWs may be very high following the actual pandemic [8]. COVID-19 is associated with insufficient staff, and exhaustion may increase sick leave, turnover [49], and even the resignation of HCWs [32,34]. The lack of HCWs associated with psychological distress may place health systems from multiple countries at risk [50,51]. The organizational structure and function of essential health services will have to challenge the population’s health needs [50]. Although all countries experienced the COVID-19 pandemic, the impact differed depending on the integrity, resiliency, and capacities of the health systems [51]. Preventive strategies for the mental health of HCWs are urgently needed [31,38].

### 4.2. Long-Term PTSD

We demonstrated a greater prevalence of PTSD (19%) within the six months after the end of the SARS-CoV-1 epidemic, with still preoccupying long-term prevalence (8%) despite a decrease after one year [34,38]. No study has evaluated PTSD more than three years after the end of the epidemic [13,40]. A prevalence of PTSD up to 44% as reported in SARS patients four years after their infection [52], and the prevalence may remain high even decades after a traumatic event [53]. PTSD could become a lifetime disorder [1] in some HCWs. Considering the very long-term consequences of PTSD and considering that the recurrence of a traumatic event can worsen existing PTSD [13], workplace interventions should focus on the prevention of PTSD in HCWs. Some of the symptoms could be managed with proper treatment and psychological help [1,36]. With appropriate training, HCWs can improve their coping strategy with a better understanding of the transmission process [13,31]. Interestingly, one study described lower numbers of PTSD in HCWs working in low-contact wards, but PTSD became considerably more symptomatic with time [31]. HCWs from low-contact wards had more intrusive thoughts after the epidemic, maybe because they did not cope during the epidemic [31].

### 4.3. Contact with Patient with SARS: A Risk Factor for PTSD

Although never demonstrated in HCWs in contact with SARS-CoV-1 patients, it seems that dose–response can be applied to PTSD [23,24]. However, despite a prevalence of PTSD at 22% in HCWs in high-contact SARS-CoV-1 wards and a prevalence at 9% for those working in a low-contact ward, we failed to demonstrate a significantly higher proportion of PTSD. Only one study demonstrated a greater prevalence of PTSD in HCWs directly in contact with SARS-CoV-1 patients [38]. In line with the literature, several hypotheses have been proposed for a higher psychological impact on those HCWs. HCWs in SARS wards may have fear for themselves and their families since they could bring home the virus [30,32]. The media has put pressure on HCWs working in SARS wards, describing them as “Heroes” but also as “infection vectors” [32]. HCWs could also become patients increasing stress and causing a psychological stigma such as was demonstrated by the Ebola epidemic [11]. Moreover, HCWs in SARS wards are often understaffed, meaning that the workers are working more hours and taking care of more patients than they usually do during a health crisis, increasing work stress and anxiety [32]. Facemasks and other protective measures could also be perceived as hard to work with, increasing difficulties in communication and socializing [15,38,39]. Contrary to this, such protective measures were already part of the working environment for some HCWs, such as emergency or intensive care unit workers [31,33], which may, therefore, have attenuated their stress of the epidemic. The prevalence of PTSD between physicians, nurses, and other caregivers was not assessed. Identifying caregivers as the most at risk of PTSD, as well as determining the factors influencing the risk of PTSD, are necessary [35,39,54].

### 4.4. Other Influencing Factors for PTSD

In addition to the the time effects, the only other significant factor influencing the prevalence of PTSD was the number of children. Two hypotheses may explain the increased risk of PTSD for HCWs with children. Some HCWs may have intrusive thoughts of the fear of spreading the virus to their families and children, which may lead to avoidance behavior [37,42,55]. Moreover, as previously demonstrated for other outcomes, HCWs are mostly women, and they still have more at-home responsibilities than men in most countries [56]. The education of children, nursing, and household care can be hard to combine with a full-time job [42]. Even if we did not demonstrate that women were more at risk than male HCWs, women might be more at risk of PTSD [1,2], caused by the additional strain imposed because of their social roles [42]. Some studies showed that young adults may be at higher risk of PTSD [57]. Lack of experience [34,38] may be compensated by organizing teamwork between the older and younger HCWs [31]. Because of lacking data, meta-regressions on education levels were impossible. However, some studies showed that HCWs with less education might be more at risk of PTSD, putatively in relation to their awareness of the situation [13,14]. Being married showed contradictory results in the literature: it may help HCWs to cope with the stress of epidemics [35], but it may also increase the fear to infecting their loved ones, and thus the risk of PTSD [13,14]. Personality could be a risk factor for PTSD [54]. HCWs with psychiatric backgrounds might be more at risk since their psyche is already labile [34]. An interesting factor that can influence the risk of PTSD in HCWs could also be their willingness to be in contact with the virus [31].

### 4.5. Limitations

Our study has some limitations. Meta-analyses inherit the limitations of the individual studies of which they are composed: the varying quality of studies and multiple variations in study protocols and evaluation [55,58]. The literature search and selection procedure may have introduced biases since only studies written in English were used. However, the use of broader keywords in the search strategy limits the number of missing studies [59]. We conducted the meta-analyses on only published articles; therefore, they were theoretically exposed to publication bias. The sample size may seem low, but the first epidemic of SARS-CoV-1 was geographically localized [12]. Consequently, the generalizability of our results is deemed impossible, as the first SARS-CoV-1 epidemic was mostly in South East Asia and in Canada [12]. All of the included studies were based on self-reported questionnaires that may lack standardized interviews; however, self-report questionnaires enable comparisons between the studies. The percentage of respondents within those studies may seem low, from 15% [37] to 75% [39]; however, the response rate was higher than usual [60,61]. Questionnaires are also subject to declarative bias [62]. Even if almost all of the studies used the same scale (IES/IES-R), different scores were used for the diagnosis of PTSD that may explain the high heterogeneity (I^2^ > 80%) between the studies for the prevalence of PTSD and the low heterogeneity (I^2^ < 15%) fof the IES scores. Only one study questioned the HCWs about previous psychiatric disorders [34], and our meta-analyses may lack control for confounding factors. As most HCWs are female, selection bias is more likely to arise. The data collection and inclusion/exclusion criteria were similar but not identical between the studies, which may have affected our results. Almost all studies were cross-sectional, giving providing an image at a given time. Some studies did not explicitly report an ethical approval; however, the rate is similar to some other previous meta-analyses [55,63,64]. The lack of data on the long-term prevalence of PTSD suggests implementing cohort studies for several years to assess the psychological impacts in the long term. The lessons from the SARS-CoV-1 epidemic may be useful in building preventive strategies to detect the HCWs at risk and help them faster. In this context, occupational health services, health executives, and hospital managers should be aware of the risk.

## 5. Conclusions

The HCWs had a high psychological impact during the first epidemic of SARS-CoV in 2003. The prevalence of PTSD in HCWs was high (21%) during the first epidemic and remained high in the long term (13% more than one year after the end of the epidemic). PTSD in HCWs seemed to be higher in HCWs working in departments with high contact with SARS-CoV-1. Younger HCWs and female HCWs seemed the most impacted by PTSD. The lessons learned from the SARS-CoV-1 epidemic may help prevent a public health issue in HCWs, with a wave of PTSD that may follow the recent COVID-19 pandemic.

## Figures and Tables

**Figure 1 ijerph-19-13069-f001:**
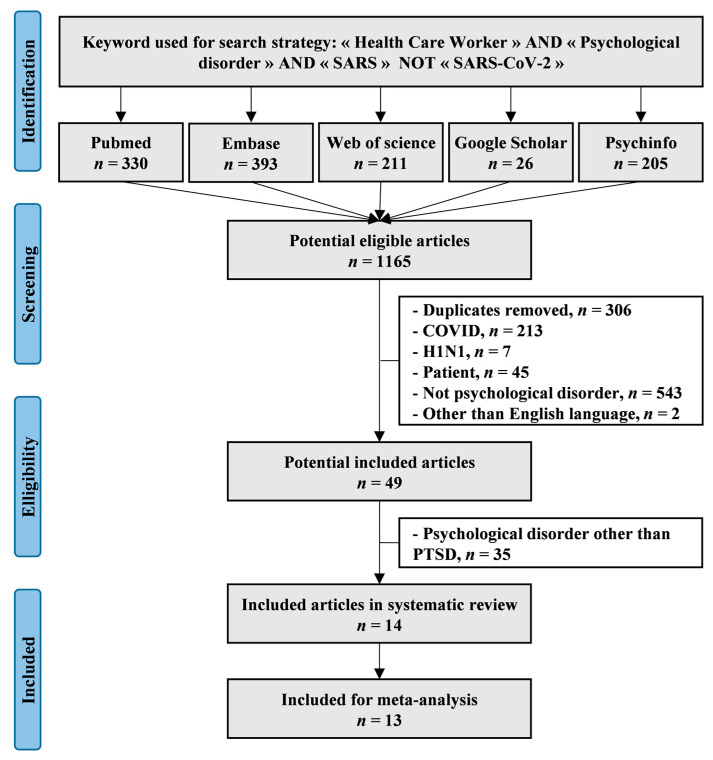
Flow chart, i.e., flow diagram of study selection following the Preferred Reporting Items for Systematic Reviews and Meta-Analyses (PRISMA).

**Figure 2 ijerph-19-13069-f002:**
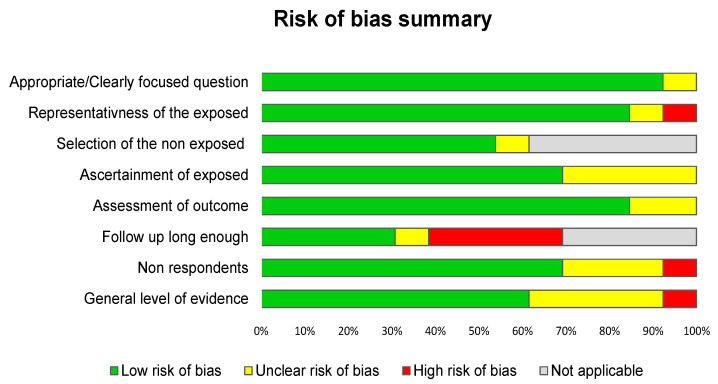
Summary of risk of bias using the Scottish Intercollegiate Guidelines Network (SIGN).

**Figure 3 ijerph-19-13069-f003:**
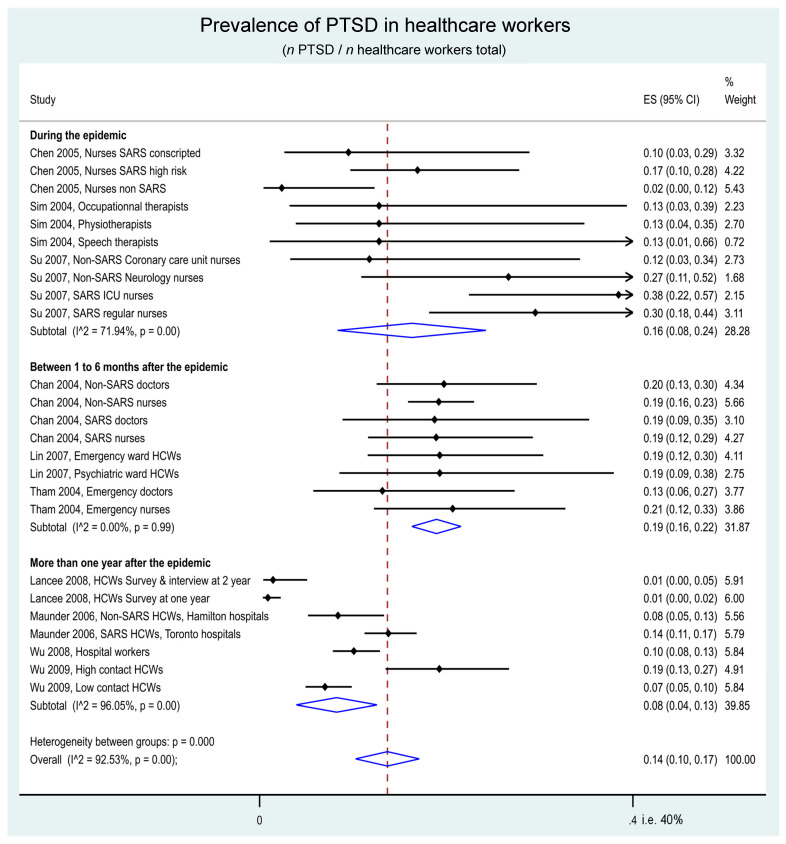
Meta-analysis on prevalence of post-traumatic stress disorder following the SARS-CoV-1 epidemic of 2003, by period of follow-up.

**Figure 4 ijerph-19-13069-f004:**
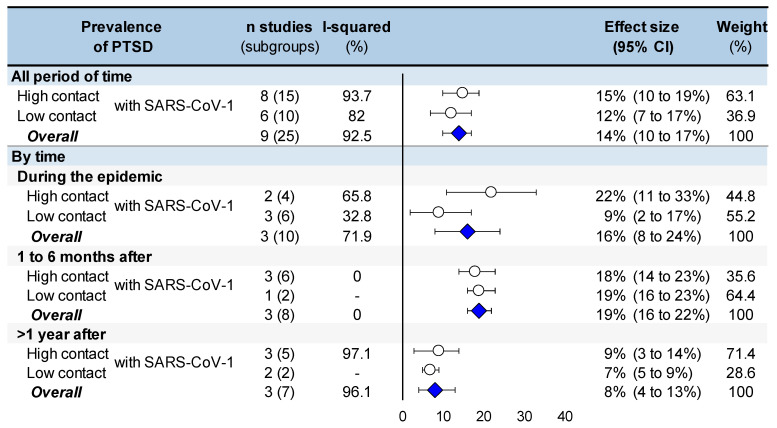
Summary of meta-analysis on prevalence of post-traumatic stress disorder following the SARS-CoV-1 epidemic of 2003, by contact with SARS-CoV-1 (high or low contact) and by time (follow-up). Each summary of meta-analysis is represented in the forest-plot by a white circle (dot) on a horizontal line. The dots represent the pooled-effect estimate (i.e., prevalence in this case), and the length of each line around the dots represent their 95% confidence interval (95CI). An overall summary of the results of meta-analyses is represented by a blue lozenge below.

**Figure 5 ijerph-19-13069-f005:**
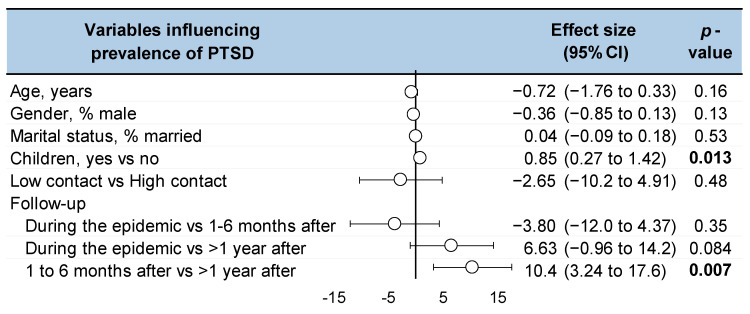
Factors influencing (meta-regressions) the prevalence of post-traumatic stress disorder following the SARS-CoV-1 epidemic of 2003. Each summary of meta-analysis is represented in the forest-plot by a white circle (dot) on a horizontal line.

**Table 1 ijerph-19-13069-t001:** Characteristics of included studies. DTS-C: Davidson Trauma Scale-Chinese version; ED: Emergency department; ICU: Intensive Care Unit; IES/IES-R: Impact Event Scale/Impact Event Scale Revised.

Study	Country	Design	Category of	Contact	Hospital	n	Age	Gender	PTSD		Measurement
			HCWs	High/Low	Department		Years	% Men	n	Scale	Time
**Chan 2004**	Singapour	Cross- sectionnal	Physicians/Nurses	High	SARS unit	106	n per age group	-	20	IES	2 months after epidemic
			Physicians/Nurses	Low	Non-SARS unit	555			107		
**Chen 2005**	Taiwan	Cross- sectionnal	Nurses	High	SARS unit	86	26.5 ± 3.1	0	-	IES	During the epidemic
				Low	Non-SARS unit	42					
**Chong 2004**	Taiwan	Cohort	All	High/Low		1257	31.8 ± 6.43	18.9	-	IES	
**Lancee 2008**	Canada	Cohort	All	High	ED, SARS unit, ICU	448	41.3 ± 10.2	14	4	IES	1 year after epidemic
						139	45.0 ± 9.6	13	2		2 years after epidemic
**Lin 2007**	Taiwan	Cross- sectionnal	All	High	ED	66	33.5	7.6	13	DTS-C	1 month after epidemic
				Low	Psychiatry	26	34.5	11.5	5		
**Lu 2006 ***	Taiwan	Cross- sectionnal	Physicians			24	36.5 ± 6.7	100			
			Nurses	High	SARS unit	49	31.6 ± 5.5	6.1	-	-	During the epidemic
			Other HCWs			54	31.1 ± 7.6	48.1			
**Maunder 2006**	Canada	Cross- sectionnal	All	High	SARS unit	587	n per age group	14	81	IES	1 year after epidemic
				Low	Non-SARS unit	182		10.4	15		
**McAlonan 2007**	Hong-Kong	Cohort	All	High		106	n per age group	29.2	-	-	During The epidemic
				Low		70		24.3			
				High		71		33.8		IES-R	1 year after epidemic
				Low		113		37.1			
**Sim 2004**	Singapour	Cross- sectionnal	Physiotherapist	Low	Rehabilitation unit	18	n per age group	-	2	IES	
			Occupational therapist			13			2		During the epidemic
			Speech therapist			3			1		
**Su 2007**	Taiwan	Cohort	Nurses	High	SARS ICU	26	31.5 ± 6.2	0	10	DTS-C	During the epidemic
					Regular SARS unit	44	29.8 ± 7.6		13		
				Low	ICU	17	32.7 ± 4.3		2		
					Neurology	15	25.4 ± 3.7		4		
**Tham 2004**	Singapour	Cross- sectionnal	Physicians	High	Emergency	38	31.6 ± 4.4	65.8	5	IES	6 months after epidemic
			Nurses			58	32.1 ± 9.2	8.6	12		
**Verma 2004**	Singapour	Cross- sectionnal	General practitionner	High		32	45.0 ± 11.2	60.6	-	IES	During the epidemic
				Low		682					
			Traditionnal Chinese Medicine	High		1	50.1 ± 9.0	59			
				Low		326					
**Wu 2008**	China	Cross- sectionnal	All	High		549	n per age group	23.5	55	IES	3 years after epidemic
**Wu 2009**	China	Cross- sectionnal	All	High	SARS unit	135	n per age group	23.5	26	IES-R	3 years after epidemic
				Low	Non-SARS unit	414			29		

*: included in the systematic review but not in the meta-analysis.

## Data Availability

All information are available in the manuscript.

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
