# Peer review of "Prevalence of Post-Traumatic Stress Disorder (PTSD) in Healthcare Workers following the First SARS-CoV Epidemic of 2003: A Systematic Review and Meta-Analysis"

_ijerph, 2022, doi:10.3390/ijerph192013069_

Round 1
Reviewer 1 Report
The authors have performed a quality study. However, it can be said that the contribution of this study to the literature is limited. In the introduction, it is not clear which gap this study will fill in the literature and its contribution to the literature.
There are study results revealing that healthcare workers suffer from post-traumatic stress disorder due to the new coronavirus, which has been on the agenda for the last 2-3 years. It is already known that this PTSD is likely to occur to healthcare workers after such public health emergencies. I do not think that revealing the prevalence of PTSD and related factors after the epidemic that emerged 10 years ago will contribute.
Author Response
Dear reviewer, thank you for your comment. Our response is in the word/pdf.
Kind regards

Reviewer 2 Report
Hello,
Thanks for allowing me to read your valuable work. In general, the manuscript would benefit greatly from an English language editor as many prepositions are missing and the grammar could greatly be improved in the introduction, the results, and the discussion. It seems like the manuscript had two major writers, one for the introduction and the discussion, and one for the methods and results. A more coherent writing style would make the manuscript more readable.
Abstract:
1. Examples The first sentence should read:"Past epidemics not past epidemic. This is just one of many examples.
2. The abstract needs to establish the need for the study.
3. Given that we just went through another pandemic, why was SARS-COVID1 chosen for this work?
4. It is interesting that lessons are mentioned in the abstract but then never again.
Introduction
1. Please define PTSD, it is not a disease it is a disorder.
2. What is the baseline rate of PTSD in HCW without SARS-COv? Only then are we able to see if this is indeed a significant increase in this population?
3. Please explain the meaning of this sentence:" The occurrence of PTSD increases over the first year following a stressful event. For whom? why? what is the mechanism?
4. This sounds like speculation:" Although never demonstrated in HCWs in contact with SARS-CoV patients, it seems that dose-response can be applied to PTSD." That may be good for the discussion section but is inappropriate in the introduction, we want to know what is the current state of the knowledge and the gaps, and hopefully what this meta-analysis will contribute to closing some of those gaps.
5. "However, HCWs the most at risk of PTSD during an epidemic is not well known. "This seems to be the gap, but you later do not pick up on this issue again. Do you mean, which type of HCWs, or which demographic factors impact all HCWs to develop PTSD? This is not clear at all.
Figure 1
1. What do psychological matters mean, the same as psychological disorders?
Statistical considerations
1. Please provide a reference for the pooled random-effects meta-analyses and explain briefly its benefits to other forms of meta-analyses.
2. You stratified your data based on what criteria? Does PTSD change at 1 month, six months, and thereafter?
3. Please explain if you want high or low heterogeneity as a quality indicator.
4. Can you please explain why this was important to do? : " Finally, we repeated all meta-analyses and meta-regressions on scores of PTSD, for each dimension of the IES: IES-total, IES-intrusive, and IES-avoidance."
Results
1. I am concerned you included four studies without ethical approval. Did you contact the authors about that?
Table 1
1. Please exchange doctor for the term physician. Doctor is a title, physician a profession. What is a general practitioner?
2. What does n per age group stand for?
3. You have no legend for Table 1 to explain IES, IES-R.......
Figure 2
Figure 2. Summary of risk of bias.(SIGN. what does SIGN stand for?
2. How do you know this is true if four studies did not mention IRB approval:" All included studies focused on HCWs and were conducted on a voluntary basis." That is how the sentence should read.
3. Please replace sex for gender.
4. All studies reported the prevalence of HCWs with PTSD, how did they report PTSD in HCWs, as a baseline assessment score? I am confused by this sentence.
Figure 3 is too small and really hard to read. What do the 0 and 4 on the y-axis stand for?
4. On page 7 you say:" Most I2 were high (>80%) highlighting heterogeneity between results (Figure 3 and Figure 4). On page 8 you say Most I2 were low (<15%) for IES total, IES avoidance and IES intrusive. Can you explain that?
Discussion
1. "The main findings were that the prevalence of PTSD for the first SARS epidemic in 2003 among HCWs is high, around 14%." Compared to what data do you make this statement?
2. Please explain:"Health system in multiple countries may be at risk."
At risk for what, why?
3. This statement should move to the introduction as it indicates gaps. "A previous meta-analysis was conducted on psychological consequences of the SARS-CoV- 294 1 but did not assess the prevalence of PTSD and did not study putative influencing factors such as contact with virus, time effect, or the importance of some sociodemographic."
4. The whole section on Contact with patient with SARS: a risk factor for PTSD needs to be carefully read and improved so it is not just a personal opinion piece.
References
Many of your references are in various reference styles, choose one and be consistent.
Author Response

(The authors gave the same response as above.)

Reviewer 3 Report
Please refer to my comments in the manuscript

Author Response

(The authors gave the same response as above.)

Round 2
Reviewer 1 Report
I have no further comment.
Reviewer 2 Report
Thank you for your thorough responses to my concerns. They were well addressed and I have no further comments to make.